# How Perceived Quality of Care and Job Satisfaction Are Associated with Intention to Leave the Profession in Young Nurses and Physicians

**DOI:** 10.3390/ijerph17082714

**Published:** 2020-04-15

**Authors:** Peter Koch, Max Zilezinski, Kevin Schulte, Reinhard Strametz, Albert Nienhaus, Matthias Raspe

**Affiliations:** 1Centre of Excellence for Epidemiology and Health Services Research for Healthcare Professionals (CVcare), University Medical Centre Hamburg-Eppendorf, 20246 Hamburg, Germany; Albert.Nienhaus@bgw-online.de; 2Business Division Nursing Directorate, Nursing Science, Core-Team III Delirium Management und Dementia Care Charité—University Medicine, 10117 Berlin, Germany; max.zilezinski@charite.de; 3Clinic for Internal Medicine, University Medical Centre Schleswig-Holstein, Campus Kiel, 24105 Kiel, Germany; kevin.schulte@uksh.de; 4Wiesbaden Business School, College RheinMain University of Applied Science, 65183 Wiesbaden, Germany; Reinhard.Strametz@hs-rm.de; 5Institution for Statutory Accident Insurance and Prevention in the Health and Welfare Services (BGW), 22089 Hamburg, Germany; 6Department of Internal Medicine, Infectious Diseases and Pulmonary Medicine, Charite - University Medicine, 10117 Berlin, Germany; Matthias.Raspe@charite.de

**Keywords:** quality of care, job satisfaction, intention to leave, inpatient patient care, hospital staff

## Abstract

German hospitals are now confronted with major challenges from both shortages and fluctuations in the numbers of physicians and nurses. This makes it even more important that physicians and nurses do not prematurely leave patient care. The objective of the present study was to improve our understanding of the factors that trigger intentions to leave the profession. For this purpose, data from 1060 young physicians and nurses in hospital care were analysed. Intentions to leave the profession was assessed with the Copenhagen Psychosocial Questionnaire (COPSOQ). In the first step, the association was determined between intention to leave the profession and the factors of perceived quality of care and job satisfaction. In a second step, a mediation analysis was performed to determine the effect of perceived quality of care after correction for the possible mediator of job satisfaction. There were statistically significant negative associations between perceived quality of care and intention to leave the profession (beta: −2.9, 95% CI: −4.48–−1.39) and job satisfaction and intention to leave the profession (beta: −0.5, 95% CI: −0.64–−0.44). The effect of perceived quality of care on intention to leave the profession was partially mediated by job satisfaction. Thus, high perceived quality of care and high job satisfaction are both important factors that tend to prevent young physicians and nurses from leaving their professions.

## 1. Introduction

German hospitals now face major challenges from both shortages and fluctuations in the numbers of physicians and nurses. This concerns both employers in regions with poor infrastructure and hospitals in conurbations that provide maximal care [1].

A systematic review concluded that between 21% and 54% of German physicians are considering leaving work with direct patient contact [2]. International studies have reported that important reasons for this include overtime, psychosocial stress, burnout, career aspects and particularly, a low level of job satisfaction [2,3,4,5,6]. In order to improve their balance between work and family, some physicians switch to nontherapeutic work or emigrate, in the hope of finding better working conditions in other countries [7]. 

Although the absolute number of all physicians in Germany has been constantly increasing over the years [8], it is still justified to discuss a “relative lack of physicians” or a “lack of medical working hours”. The reasons for this include the relatively high number of hospital beds (in comparison with other European countries), the imbalanced distribution between conurbations and rural areas, the fact that many physicians are very highly specialized, and finally the entry of more women into medicine [9]. At the end of 2018, it was found that women made up 47% of physicians registered at the German Medical Association [10]. This proportion is increasing, as is the proportion of female physicians who work part-time.

The position was similar for nursing staff. Even during training, 20–30% of trainees were considering not remaining in the profession for more than 5 years [11]. Systematic reviews have proposed a wide variety of reasons that trained nursing staff may consider leaving the profession. These include inadequate earnings, lack of personnel, lack of autonomy and high stress at work [12,13,14]. At the individual level, these factors lead to low job satisfaction, which is also empirically associated with thoughts of leaving the profession [15,16,17,18,19,20,21].

Moreover, for both physicians and professional nurses, there is evidence that poor quality of care is associated with low job satisfaction [22,23,24]. It is unclear whether the quality of care influences job satisfaction, or whether the converse is the case; other surveys have considered that the association is in the converse direction, or that the two factors interact [25,26].

For both professional groups, there is little evidence that poor quality of care is associated with leaving the profession, or that there is a causal relationship. However, in the European nurses early exit (NEXT) study, this association was found for nurses who had left their profession. According to this study, the two most important reasons for leaving the profession were poor quality of care and time pressure [27]. There is even less information for physicians. A multinational prospective cohort study of young physicians found an association between the frequency of situations in which the subject regretted the result of the treatment and increased intention to leave the profession [28].

In view of currently available data, the role of poor quality of care in leaving the profession, taking poor job satisfaction into consideration, is not clear. Therefore, we formulated the following research questions: Is low perceived quality of care associated with intention to leave the profession in young physicians and nurses?To what extent is an existing association between poor quality of care and intention to leave the profession mediated by job satisfaction?

This analysis of the associations for young employees could help us to improve our understanding of why members of the target group may wish to leave the profession. Appropriate reforms to improve specific conditions at work could lead to timely and specific prevention of departure from the profession. This would help to retain the necessary staff in the hospitals and guarantee that German hospitals will continue to function efficiently.

## 2. Materials and Methods

Data were taken from a large survey of young physicians and nurses active in acute hospital care and were reanalysed [29]. In September 2017, the Institution for Statutory Accident Insurance and Prevention in the Health and Welfare Services (BGW) commissioned a randomised cross-sectional study on young hospital employees throughout Germany. The cooperation partners for this project were seven medical associations or societies and a professional nursing association (Marburger Bund, Hartmannbund, German Association for Internal Medicine, Professional Association of German Internal Physicians, German Society for Paediatric and Developmental Medicine, German Society for Gynaecology and Obstetrics, German Society for Anaesthesiology and Intensive Care, German Association of Professional Nurses). The field access was based on the member databases of the participating professional associations or societies. 

As this study was an anonymous survey without sensitive personal data, no discussion or vote from the ethics committee was necessary.

The study population consisted of young physicians or nursing employees (≤35 years), who worked in hospitals and who had had a maximum of six years professional experience. Before the start of the survey, the societies or associations sent a flyer to their members. The homepage of the online survey presented essential information on the study. This included the study objective, the voluntary character of participation, the maintenance of anonymity, the conformity with the regulations of the Federal Law on Data Protection, the time for completion of the questionnaire and the consent for participation by completion of the questionnaire. The online survey was performed with the survey program EFS Survey from Questback/Unipark, with safety requirements in accordance with ISO 27001 on the basis of IT baseline security. The questionnaire was pretested with 40 participants. As incentives for participation, tickets were raffled for the Capital City Congress on Medicine and Health and the German Nursing Meeting. In all, 10,162 members (6362 physicians, 3800 nurses) were sent an email with an invitation to the online survey. Two and four weeks later, all those invited were sent an email to remind them of their participation in the study. We wished to achieve a balance between the numbers of physicians and nurses—a 1:1 ratio in the sample. We therefore planned to invite exactly the same number of physicians as the total number of nurses given by the German Association of Professional Nurses in their member database as fulfilling the inclusion criteria. Thus, data collection was complete for the nurses. In order to minimise the risk of selection bias and to select the same number of physicians in the target population, the physicians were subjected to proportional stratified randomisation, depending on the total numbers of members per society or association.

Work-related psychosocial factors were assessed with the Copenhagen Psychosocial Questionnaire (COPSOQ) [30]. Possible factors were surveyed, including working speed, work-privacy conflict, quality of leadership, presenteeism, predictability of the work and job satisfaction. These COPSOQ scales (scale: 0–100) were dichotomised using the cut-off of ≥50 points. Other adjustment variables, such as self-rated health and burnout, were also taken from the COPSOQ questionnaire.

To measure the perceived quality of care, we had recourse to the German version of the original instrument. This was developed for physicians and has been validated by Loerbroks et al. [31,32]. The corresponding scale for professional nurses was derived from the original scale, by adapting three of the six items for nursing work. The scale exhibited good internal consistency (Cronbach’s Alpha: 0.80 (physicians) and 0.77 [33]). The scale was dichotomised in accordance with Loerbroks et al., on the basis of the third tertile of the actual distribution (third tertile: 2.83) [31].

The outcome variable “intention to leave the profession” was also taken from the COPSOQ questionnaire. The variable was dichotomised as in Hasselhorn (2005), on the basis of the answers to the question (33): “In the last 12 months, how often have you thought about leaving your profession?” The five possible answers are: 0 = never/several times per year, 1 = several times a month/several times a week/every day. We also checked the influence of workplace-related characteristics, such as weekly working hours, weekend work, night shifts and employment relationship. Bivariate associations were checked with the Spearman correlation coefficient. Chi^2^ tests were calculated for nominal variables. Linear regression models were calculated for multivariate analysis (the residuals were normally distributed). The following variables were included in the multivariate model—perceived quality of care, job satisfaction, working speed, work-privacy conflict, quality of leadership, presenteeism, predictability of work, workplace-related characteristics, burnout, self-rated health, age, gender and professional group. Using the stepwise backwards method according to Hosmer and Lemeshow, variables with *p* > 0.1 were eliminated stepwise from the model [34].

Mediation analysis was performed using the 4-step procedure for cross-sectional data as described by Baron and Kenny [35]. The basic assumption of mediator analysis is the causal association between the independent variable (X) and dependent variable (Y). Three regression models were used to test the assumed relationships in the theoretical path model between X (perceived quality of care), mediator (M) (job satisfaction) and Y (intention to leave the profession). In the first step, the association was tested between X (perceived quality of care) and Y (intention to leave the profession). In the second step, the association was tested between X (perceived quality of care) and M (job satisfaction), where the mediator takes the role of a dependent variable. In the third step, the influence of M (job satisfaction) on Y (intention to leave the profession) was tested. In the last step, X (perceived quality of care) was also incorporated in the model from step 3. According to Baron and Kenny, the effect of M (job satisfaction) should then be retained. If the effect of X then drops to zero, this is rated as complete mediation. If an effect of X is retained, as is the effect of M, partial mediation can be assumed. 

The analyses were performed with the statistics program SPSS (IBM SPSS Statistics for Windows, Version 23.0. Armonk, NY: IBM Corp.) mediation analysis was performed with PROCESS, as described by Hayes [36]. To calculate the confidence intervals of the total, direct and indirect effects, bootstrapping was performed with 5000 iterations. Effects were regarded as significant if the limit of confidence did not include zero. 

## 3. Results

### 3.1. Demographic and Work-Related Characteristics of the Sample 

A total of 1337 employees took part in the online survey. The response rate was 13% (physicians 18.5%, nurses 7.5%). After applying the inclusion criteria, 1060 cases remained for evaluation (physicians: 80.7%, nurses: 19.3%). The mean age of the employees was 29.9 years. In the mean, the physicians were older than the nurses (30.8 vs. 26.5 years, *p* = 0.001) (Table 1). A total of 62% of the sample were women; this figure was greater for the nurses than for the physicians (70% vs. 60.4%, *p* = 0.013). The percentage of nursing staff who worked more than 48 h weekly was much lower for nurses than for physicians (10.3% vs. 71%, *p* < 0.001). A total of 20.6% of the nurses worked on weekends at least three times a month. This percentage was only 13.4% for physicians (*p* = 0.01). For the whole group, the 37.6% of the employees worked night shifts at least 6 times a month; there was no significant difference between nurses and physicians in this respect. The percentage with a fixed-term contract of employment was much lower for nurses than for physicians (23.3% vs. 89.4%, *p* < 0.001). The findings for the dichotomised COPSOQ scales were as follows. For nurses, the percentage with a high working speed was greater (88.0% vs. 77.4%, *p* = 0.001). On the other hand, work-privacy conflicts were rarer for the nurses (50.7% vs. 62.1%, *p* = 0.003). Low levels of quality of leadership were rarer for nurses than for physicians (59.4% vs. 68.9%, *p* = 0.01). For the whole group, the percentage of presenteeism was 46% and of low predictability (65.2%); there was no statistically significant difference between nurses and physicians in this respect. Low job satisfaction was commoner for nurses than for physicians (53.5% vs. 34.1%, *p* < 0.001), as was low perceived quality of care (58.8% vs. 31.3%, *p* < 0.001), which was almost twice as frequent as for physicians. The overall percentage of those who frequently considered leaving the profession was 30.9% for the whole sample, and was greater for nurses than for physicians (42.2% vs. 28.2%, *p* < 0.001).

### 3.2. Mediation Analysis to Investigate the Associations between Perceived Quality of Care, Job Satisfaction and Intention to Leave the Profession 

Table 2 shows the four steps of the mediation analysis. In the first step, it was found that the perceived quality of care had a statistically significant effect on intention to leave the profession (= total effect, beta: −4.3, 95% CI: −5.94–−2.72). In step 2, a statistically significant association was found between X (perceived quality of care) and the mediator (job satisfaction) (beta: 2.6, 95% CI: 1.63–3.53). The association between mediator and intention to leave the profession (step 3) was also statistically significant (beta: −0.6, 95% CI: −0.68–−0.48). The final step investigated the direct effect of perceived quality of care after correction for the mediator (beta: −2.9, 95% CI: −4.48–−1.39). For the mediator of job satisfaction, there was still a statistically significant effect on beta: −0.5 (95% CI: −0.64–−0.44). For work-privacy conflict, there was also a statistically significant effect (beta: 0.2, 95% CI: 0.10–0.26). The following values were calculated for the standardised coefficients in this model—perceived quality of care: −0.10, *p* < 0.001, job satisfaction: −0.33, *p* < 0.001, work-privacy conflict: 0.15, *p* < 0.001, professional group: −0.04, *p* = 0.245 (not listed separately in the table). In the stratified analysis by professional group, the observed estimates for this model remained statistically significant for physicians. However, for nurses, the estimates for perceived quality of care and work-privacy conflict were no longer statistically significant. 

For the indirect effect of perceived quality of care on intention to leave the profession, there was a statistically significant association of beta: −1.4, 95% CI: −2.01–−0.81 (Figure 1). Thus, the magnitude of the indirect effect corresponds to about 33% of the total effect (beta: −4.3, 95% CI: −5.94–−2.72). In summary, the data for the study group of young employees show that both the perceived quality of care (beta: −2.9) and job satisfaction (beta: −0.5) are negatively associated with intention to leave the profession. Job satisfaction acts as a partial mediator for the perceived quality of care. The indirect effect of perceived quality of care as mediated by job satisfaction corresponded to beta: −1.4. The direct effect was about twice as large (beta: −2.9).

## 4. Discussion

The objective of the present study was to investigate the influence of perceived quality of care and job satisfaction on intention to leave the profession. The subjects were young employees in hospital health care. We also investigated whether the perceived quality of care depended on job satisfaction and quantified any effect.

### 4.1. Prevalence in Young Physicians and Nurses of Low Values of Perceived Quality of Care, Low Job Satisfaction and Intention to Leave the Profession 

For this young group of subjects, prevalence values were found to be high for low perceived care, low job satisfaction and intention to leave the profession. For all three variables, the percentage of nurses was significantly greater than for physicians. For the physicians in this study, the mean value for perceived quality of care (x (SD): 2.4 (0.9)) was less favourable than the 2016 findings for German physicians in hospitals (x (SD): 1.9 (0.8)) [31]. Possibly, this difference in perceived quality of care can be explained due to selection bias. There are no comparable values with the same scale for nurses.

The German COPSOQ reference values for job satisfaction were determined for hospital employees aged up to 34 years for the period 2011 to 2016 [37]. Comparison with these reference values shows that values for job satisfaction are relatively unfavourable for both professional groups in this sample (sample vs. reference data x (SD) physicians: 58.6 (15.8) vs. 62.9 (15.6), nurses: 50.6 (16.8) vs. 59.4 (16.3)).

For the physicians in our study, the percentage with intention to leave the profession was relatively high, namely 28.2%. In a systematic review, values between 17% and 26% were reported for international studies [2]. However, a study of German physicians reported that only 14% expressed the unambiguous wish to leave the profession [38]. On the other hand, these data were published in 2006 and are therefore relatively old. The percentage for nurses (42%) is very high in comparison with the NEXT study [27], where it was reported that 18.4% of the surveyed professional nurses often thought of leaving the profession. Within other European countries in the NEXT study, higher values were only found in Italy (20.6%) and for Finnish nurses aged under 30 (26%) [18]. A systematic review of nurses in English-speaking countries found values between 6.3% and 33% [14]. These values are apparently increasing over time. Thus, more recent studies on Italian nurses found the value to be 35.5%; for German nurses in intensive care, the reported value was as high as 49.8% [39,40]. For both professional groups in the present study, the mean values were clearly higher than the COPSOQ reference values (sample vs. reference data x (SD) physicians: 26.7 (26.8) vs. 19.8 (25.9), nurses: 36.0 (26.5) vs. 22.1 (25.3)).

With regard to the comparison to COPSOQ data of employees aged up to 34 years, no changes in the mean values of job satisfaction and intention to leave the profession could be observed in a sensitivity analysis when 35-year-olds (*n* = 50) were excluded. To what extent the differences in the mean values can be explained by working conditions that possibly deteriorate over time or by a selection bias cannot be answered in this study.

### 4.2. Associations Between Perceived Quality of Care, Job Satisfaction and Intention to Leave the Profession 

In the multivariate model, both variables, perceived quality of care and job satisfaction, were statistically significant protective factors against intention to leave the profession; the effect of job satisfaction was three times greater than that of perceived quality of care (standardised coefficients: −0.33 vs. −0.10). Comparison of the components of these two variables might possibly explain the differences in the strengths of association (Table 3). The scale job satisfaction [41] summarises the degree of satisfaction on the basis of six individual terms (professional perspectives, colleagues, physical conditions at work, management, use of personal abilities, overall job satisfaction). Perceived quality of care, modified from Loerbroks [31], incorporates the frequency of specific patient-related situations during daily clinical work (neglect of patients due to high levels of stress at work, lack of comprehensive explanation of the treatment options, errors in nursing or medication, omission of a diagnostic test or nursing assessment for patient discharge, neglect of the social and personal consequences of the disease for the patient, feelings of guilt from poor personal treatment). In contrast, the construct job satisfaction covers a wide field of different working conditions. On the other hand, perceived quality of care focusses on the aspect of patient-related conditions of work. Moreover, the comparison shows that the items on perceived quality of care describe the situations much more specifically than do the items in job satisfaction.

The association between perceived quality of care and job satisfaction has been well described in a series of studies [22,23,24].

The European NEXT study has already reported that, for nurses, there is an association between low perceived quality of care and intention to leave the profession [27]. This was a survey of former professional nurses in Europe, who were asked for the decisive reason for their leaving the profession. The most frequent reasons were given as time pressure and low perceived quality of care. Other reasons included dissatisfaction with the use of their abilities, payment and problems in relationships at work. The same sequence was found for professional nurses who had changed their employer, but had remained in nursing. In a study on English nurses, those who reported frequent intention to leave the profession also exhibited increased rates of care-related problems [42]. On the other hand, a recently published study of Italian nurses found no association between perceived quality of care and intention to leave the profession [39]. There have hardly been any studies on physicians on the association between perceived quality of care and intention to leave the profession. Preliminary evidence is provided by the above mentioned multinational prospective cohort of young physicians, in which there was an association between frequent situations in which the subjects regretted the result of the treatment and increased thoughts of leaving the profession [28].

The association between low job satisfaction and intention to leave the profession has been described in several studies, both for physicians [2,3,4,5,6] and nurses [15,16,17,18,19,20,21,43]. A recent study on physicians in Saxony found an association between low job satisfaction and the tendency to move to other health systems [44].

### 4.3. Association in the Mediation Analysis Between Perceived Quality of Care, Job Satisfaction and Intention to Leave the Profession 

The mediation analysis showed that one third of the effect of perceived quality of care is indirectly mediated by job satisfaction (partial mediation). Two thirds of the total effect of perceived quality of care then directly affect intention to leave the profession. The effect of partial mediation may be explained as follows—the scale perceived quality of care assesses the frequency of patient-related situations that are undesirable from the professional point of view. As these critical situations become more frequent, the employees’ perceived job satisfaction decreases. Reduced job satisfaction then leads to intention to leave the profession. This association has also been empirically demonstrated [15,16,17,18,19,20,21]. However, it is just as conceivable that low perceived quality of work during the normal working day may lead directly to intention to leave the profession. It may be the case that conditions at work that prevent employees in social work from implementing their professionalism and idealism will have a negative effect on the perceived quality of care and could directly lead to thoughts of leaving the profession. There is currently lively and critical discussion of these factors, including dominating economic or social factors that influence medical, ethical or social decision in health care [45,46]. 

Both these factors, perceived quality of care and job satisfaction, are important in preventing employees from leaving the profession. Specific objectives to improve the perceived quality of care should be specifically derived from the individual items; specific measures can then be developed. There have been many different proposals for improving the care situation. It is often the case that several parties benefit simultaneously from the individual measures. Measures to improve job satisfaction, as compiled in the present study, are also of great importance, particularly in the context of the various subsidiary aspects of job satisfaction. On the other hand, these items are more likely to reflect general areas of job satisfaction. More precise statements would make it easier to formulate the required measures. For example, a study on Canadian nurses showed that good leadership was positively associated with quality of care and negatively associated with the intention to change the place of work or to leave nursing [47]. The so-called “transformational” leadership style can provide a role model for the employees. This can enhance their intrinsic motivation and help them to attain joint long-term and unselfish higher-order objectives, including improved performance [48]. In this context, another important point would be to improve the framework for inpatient care. Younger employees are demanding structural changes that could reduce the concentration of work and documentation work, which would have to be supported by adequate levels of staffing [29]. These changes could provide the foundation for job satisfaction, supported by high quality of care.

### 4.4. Association Between Work-Privacy Conflicts and Intention to Leave the Profession 

Another important finding of the analysis is the positive association between work-privacy conflicts and intention to leave the profession (beta: 0.2, *p* < 0.001). This association is consistent with the results of a few studies on social workers and teachers, European teachers, carers for the handicapped and a sample of a wide variety of professions [41,49,50,51]. These studies also found statistically significant associations between work-privacy conflict and job satisfaction [49,50,51,52]. We know of no other studies with hospital employees. 

### 4.5. Limitations

This cross-sectional study does not permit any conclusions about causality. The findings would have to be confirmed in longitudinal studies. Moreover, the data for the analysis were taken from a study with a low response rate, imbalances in the sample (ratio of 4:1 between physicians and nurses) and the possibility of a selection bias towards participation by dissatisfied young employees. In the stratified analysis by occupational group, the observed effects of the final model remain consistent for physician but not for nurses. This finding reduces the transferability of the results to nursing staff. There may also be bias in the selection of nurses and physicians, as these were restricted to members of professional or specialist societies and may therefore only represent a specific subset of physicians or nurses. These characteristics limit the representativeness of the sample.

## 5. Conclusions

The results of this study make it clear that, for young hospital employees, both perceived quality of care and job satisfaction are protective factors for intention to leave the profession. Perceived quality of care is mainly directly associated with intention to leave the profession, although there is also partial association mediated through job satisfaction. Further studies in longitudinal design are needed to confirm these findings.

## Figures and Tables

**Figure 1 ijerph-17-02714-f001:**
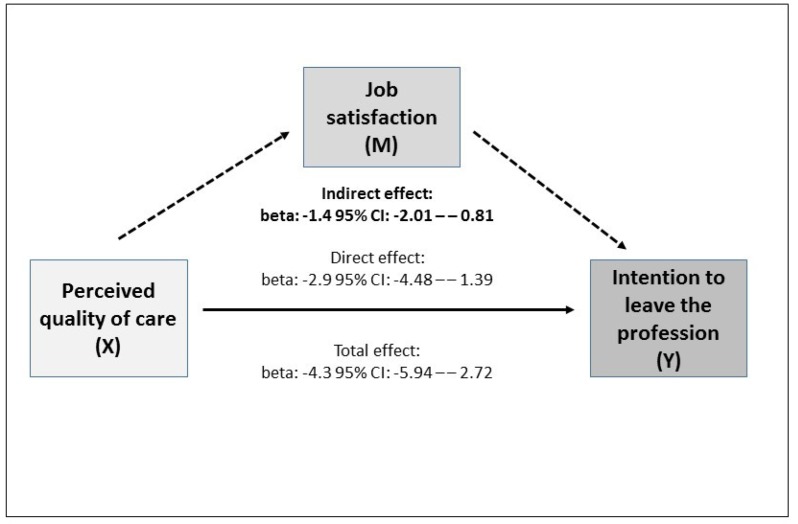
Results of the mediation analysis with portrayal of the calculated effects.

**Table 1 ijerph-17-02714-t001:** Demographic and work-related characteristics of the sample.

Characteristic	Nurses *n* = 205 (19.3%)	Physicians *n* = 855 (80.7%)	Overall *n* = 1060 (100%)	*p*
	%, x (SD*)	%, x (SD)	%, x (SD)	
Age	26.5 (3.1)	30.8 (2.4)	29.9 (3.0)	0.001
Female	70.0%	60.4%	62.2% (659)	0.013
Actual weekly working hours: ≥48 h	10.3%	71%	59.3% (626)	<0.001
Weekend work/month: ≥3	20.6%	13.4%	14.8% (156)	0.010
Night shifts/month: ≥6	39.4%	37.2%	37.6% (394)	0.552
Contract of employment: limited	23.3%	89.4%	76.7% (807)	<0.001
Working speed: high (≥50 points)	88.0%	77.4%	79.4% (829)	0.001
Work-privacy conflict: high (≥50 points)	50.7%	62.1%	59.9% (635)	0.003
Quality of leadership: low (<50 points)	59.4%	68.9%	67.1% (699)	0.010
Presenteeism: high (≥50 points)	48.3%	45.5%	46.0% (486)	0.468
Predictability: low (<50 points)	62.0%	66.0%	65.2% (688)	0.275
Job satisfaction: low (<50 points) x (SD) scale 1–100	53.5% 50.6 (16.8)	34.1% 58.6 (15.8)	37.8% (397) 57.1 (16.3)	<0.001 <0.001
Perceived quality of care: low (third tertile) x (SD) scale: 1 (high)- 5(low)	58.8% 3.0 (0.8)	31.3% 2.4 (0.9)	36.5% (373) 2.5 (0.9)	<0.001 <0.001
Intention to leave the profession: high (≥ several times a month) x (SD) scale: 1 (low)- 5 (high)	42.2% 36.0 (26.5)	28.2% 26.7 (26.8)	30.9% (326) 28.4 (27.0)	<0.001

* SD: Standard deviation.

**Table 2 ijerph-17-02714-t002:** Regression models of the mediation analysis according to Byron and Kenny.

	Intention to Leave the Profession ^1^		Job Satisfaction ^1^		Intention to Leave the Profession ^1^		Intention to Leave the Profession ^1^	
	Step 1		Step 2		Step 3		Step 4	
	beta	95% CI	beta	95% CI	beta	95% CI	beta	95% CI
Constants	18.6	−0.08–37.22	63.8**	52.80–74.80	42.7 **	24.58–60.82	53.1 **	34.25–71.86
Perceived quality of care Scale: 1 (low)–5 (high) ^2^	−4.3 **	−5.94–−2.72	2.6 **	1.63–3.53	-	-	−2.9 **	−4.48–−1.39
Job satisfaction Scale: 1–100	-	-	NA	NA	−0.6 **	−0.68–−0.48	−0.5 **	−0.64–−0.44
Work-privacy conflict Scale: 1–100	0.3 **	0.22–0.38	−0.2 **	−0.27–−0.18	0.2 **	0.12–0.27	0.2 **	0.10–0.26
Professional group (nurses vs. physicians)	−7.1 *	−11.59–−2.73	8.5 **	5.93–11.16	−4.4	−8.54–−0.21	−2.5	−6.83–1.75
R^2^	0.32		0.36		0.39		0.39	
*n*	1015		1008		1040		1006	

^1^ Adjusted for age, gender, self-rated health and burnout; ^2^ In order to facilitate interpretation, the converse scale is used; * *p* < 0.05 and ** *p* < 0.001.

**Table 3 ijerph-17-02714-t003:** Comparison of the constructs job satisfaction and perceived quality of care.

Construct	Item	Scale
Job satisfaction, COPSOQ [30]	Regarding your work in general. How pleased are you with…	5-step Likert scale
	-your work prospects?	
	-the people you work with?	
	-the physical working conditions?	
	-the way your department is run?	
	-the way your department is run?	
	-your job as a whole, everything taken into consideration?	
Perceived quality of care, modified according to Loerbroks [31]	In response to our team’s high volume of work, I have either discharged patients (for physicians) or neglected their care (for nurses).	5 step Likert scale
	I have not fully explained treatment options or answered patients‘questions.	
	I have made mistakes in treatment or medication (for physicians) or nursing care or medication (for nurses), which were not associated with my own lack of knowledge or experience.	
	I have omitted a diagnostic test (for physicians) or nursing assessment (for nurses), as I wished to discharge the patient.	
	I have hardly paid any attention to how the disease might have social or personal consequences for the patient.	
	I feel guilty, because I have treated a patient badly as a person.	

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
