# Peer review of "How Perceived Quality of Care and Job Satisfaction Are Associated with Intention to Leave the Profession in Young Nurses and Physicians"

_ijerph, 2020, doi:10.3390/ijerph17082714_

Round 1

Reviewer 1 Report

  • Summary of the manuscript

The purpose of this manuscript is to investigate the relationship between perceived quality of care and intention to leave the profession with job satisfaction as a mediator. By collecting and analyzing data from 1060 young physicians and nurses, the authors found that both high perceived quality of care and high job satisfaction are important factor in hindering young physicians and nurses from leaving the profession.

  • General comments
  • This manuscript is generally well-written, but there exist some parts to be revised and improved.
  • The dataset looks unbalanced in the numbers of physicians and nurses. Is it justifiable in the literature? Or It is inevitable in practice? As a reviewer, I wonder whether the results still hold if the same analyses are conducted for separated samples of physicians and nurses.
  • The item “Perceived Quality of Care” took an inverse scale. Is it an intentional treatment?
  • Minor Comments
  • The line 26 to 27 in Abstract can be rewritten in order to provide a clear meaning of the sentence. As a reviewer, I’m worried that some readers may wonder “what the ‘both’ cares for”. Also, the parentheses are not paired or closed. Abstract should be written in a clearer way.

Author Response

REVIEW ijerph-765740

Dear reviewer, dear reviewer,

Thank you for reviewing our manuscript and for your suggestions for improvement. We appreciate your constructive and accurate review of our manuscript.

In the following, we have individually commented on each of your points. The text has been reviewed by another native speaker and found to be good.

If you have any further questions, please do not hesitate contacting us.

With kind regards,

  1. Raspe and P. Koch representing all authors

Reviewer 1:

The dataset looks unbalanced in the numbers of physicians and nurses. Is it justifiable in the literature? Or It is inevitable in practice? As a reviewer, I wonder whether the results still hold if the same analyses are conducted for separated samples of physicians and nurses.

Authors:

Thank you for addressing this important point, which limits the external validity of the study, as the target population of this study is hospital staff in German hospitals, consisting out of a concrete ratio of physicians and nurses.

In order to achieve a more balanced ratio of nurses and doctors (1:1), it was planned in advance to invite just as many doctors as the total number of nurses of the German Professional Association for Nursing Professions (DBfK) showed, who, according to the member database, fulfilled the inclusion criteria ≤ 35 years and stationary care. A survey was therefore carried out for all members of the DBfK (N=3800).

Due to the overwhelming number of physicians in the medical societies, a randomized sample was drawn only here among them.

Randomisation was carried out at the study centre using consecutive numbers via the randomisation function in SPSS. The lists were then passed on to the supervisors of the member databases, who in turn invited the selected members to participate. In one medical society, all members who fulfilled the inclusion criteria were erroneously invited, resulting in an unbalanced ratio to the disadvantage of nurses. Additionally the response rate of nurses was very low in comparison to the one of the physicians.

A subsequent reduction in the number of participating physicians was waived for statistical reasons.

So, in fact this an unbalanced and unrealistic ratio of physicians and nurses in German hospitals (4:1). We mentioned this point in the limitation section.

In the results section it is mentioned, that in the stratified analysis the observed effects of the final model remain consistent for physicians but not for nurses.

We now added this point additionally to the limitation section as follows:

In a stratified analysis by occupational group the observed effects of the final model remain consistent for physician but not for nurses. This finding reduces the transferability of the results to nursing staff.

Reviewer 1:

The item “Perceived Quality of Care” took an inverse scale. Is it an intentional treatment?

Authors:

This is right. In table 1 we presented the data on perceived quality of care in the proper way according to the original instrument. Due to reasons of interpretation, we presented in table 2 the scale in an inverse way. We documented this and the reason in a footnote of table 2.

Reviewer 1:

The line 26 to 27 in Abstract can be rewritten in order to provide a clear meaning of the sentence. As a reviewer, I’m worried that some readers may wonder “what the ‘both’ cares for”. Also, the parentheses are not paired or closed. Abstract should be written in a clearer way.

Authors:

Thank you for this suggestion. To make it clearer we changed the sentence as follows:

There were statistically significant negative associations between perceived quality of care and intention to leave the profession (beta: -2.9, 95% CI: -4.48- -1.39) and job satisfaction and intention to leave the profession (beta: -0.5, 95% CI: -0.64- -0.44).

Reviewer 2 Report

REVIEW ijerph-765740

Thank you for the opportunity to review this manuscript. Overall, this is an important topic where the objective was to assess “what extent low perceived quality of care, low job satisfaction and intention to leave the profession are associated”. A limitation of this study it is the authors have published other manuscript in Germany (that we can not read) to Know how different they are. If this manuscript contains redundant information.

Raspe, M.; Koch, P.; Zilezinski, M.; Schulte, K.; Bitzinger, D.; Gaiser, U.; Hammerschmidt, A.; Köhnlein, R.; Puppe, J.; Tress, F. Arbeitsbedingungen und Gesundheitszustand junger Ärzte und professionell Pflegender in deutschen Krankenhäusern. Bundesgesundheitsblatt-Gesundheitsforschung-Gesundheitsschutz 2019, 1-9.

The authors refers in this manuscript: lines 71-74: As a first step, data taken from a large survey of young physicians and nurses active in acute hospital care was re-analysed [29], in order to establish whether low perceived quality of care influences thoughts of leaving the profession.

Here are some suggestions to improve the manuscript:

ABSTRACT

Please, introduce the measurement instruments or questionnaires that you have used to measure the main variables.

INTRODUCTION:

In the introduction section, please differentiate your hypotheses of the study from the objective, in order to make them clear to the readers.

The Lines 78- 81 “This analysis of the associations for young employees could help us to improve …..will continue to function efficiently” should be in the Discussion section, as a practical implications.

Please, finish the introduction section before the methodology, with the aim of the study.

MATERIALS AND METHODS

I don´t understand how was a randomised cross-sectional study? You say: “In all, 10162 members (6362 physicians, 3800 nurses) were sent an email with an invitation to the online survey”. Please, consider to expand your description of sample accordingly the power analysis used to determine the number of potential individuals that you needed to include in your study.

Was this sample, representative of the population you wanted to reach? If not, you will have to write it in the limitations section.

Please, describe what is meaning “COPSOQ”

I don’t Know how is “tempo at work”

Maybe, Table 3. Comparison of the constructs job satisfaction and perceived quality of care. Can be more usefull in the materials and methods section.

Figure 1 can be omitted, since the model with its correlations is later offered in Figure 2, which is more complete.

DISCUSSION

Do not repeat your own results (numbers) in the discussion if not absolutely necessary to make a certain point. Use descriptive language instead. For example (lines 220-225: “For this young group of subjects, prevalence values were found to be high for low perceived care (36.5%), low job satisfaction (37.8%) and intention to leave the profession (30.9%). For all three variables, the percentage of nurses was significantly greater than for physicians. For the physicians in this study, the mean value for perceived quality of care (x Ě… (SD): 2.4 (0.9)) was less favourable than the 2016 findings for German physicians with their own practices or in hospitals (x Ě… (SD): 1.9 (0.8)) [31].There are no comparable values with the same scale for nurses.

The discussion fails to clearly attempt to identify/explain reasons for results in this study that differ from other studies looking at similar outcomes.

CONCLUSION. The conclusion should state your findings. The last sentences are not conclusions of your research.

REFERENCES. Please review some mistakes in the references.

The numbers of the final references do not match the references in the text. See the references section (line 351):

  1. References
  2. Karagiannidis, C.; Janssens, U.; Krakau, M.; Windisch, W.; Welte, T.; Busse, R. Pflege: Deutsche Krankenhäuser verlieren ihre Zukunft. Dtsch Arztebl 2020, 117, A 131-133.
  3. Degen, C.; Li, J.; Angerer, P. Physicians' intention to leave direct patient care: An integrative review. Human resources for health 2015, 13, 74.

Good  luck!!!

Author Response

REVIEW ijerph-765740

Dear reviewer, dear reviewer,

Thank you for reviewing our manuscript and for your suggestions for improvement. We appreciate your constructive and accurate review of our manuscript.

In the following, we have individually commented on each of your points. The text has been reviewed by another native speaker and found to be good

If you have any further questions, please do not hesitate contacting us.

With kind regards,

  1. Raspe and P. Koch representing all authors

Reviewer 2:

A limitation of this study it is the authors have published other manuscript in Germany (that we can not read) to Know how different they are. If this manuscript contains redundant information.

Raspe, M.; Koch, P.; Zilezinski, M.; Schulte, K.; Bitzinger, D.; Gaiser, U.; Hammerschmidt, A.; Köhnlein, R.; Puppe, J.; Tress, F. Arbeitsbedingungen und Gesundheitszustand junger Ärzte und professionell Pflegender in deutschen Krankenhäusern. Bundesgesundheitsblatt-Gesundheitsforschung-Gesundheitsschutz 2019, 1-9.

The authors refers in this manuscript: lines 71-74: As a first step, data taken from a large survey of young physicians and nurses active in acute hospital care was re-analysed [29], in order to establish whether low perceived quality of care influences thoughts of leaving the profession.

Authors:

Thank you very much for this comment.

The availability of further information on this study is of course an important point to review this manuscript properly. In the cited publication we present data of this investigation on current stress factors, their consequences and subjective measures to improve the working conditions of physicians and nurses. For professional political reasons, we decided to publish these results in a German journal to address all involved parties of this situation in Germany. There are no redundant data presented in this manuscript.

The English abstract of this article can be read under the following URL:

https://link.springer.com/article/10.1007/s00103-019-03057-y

With this manuscript we want to present our findings to an international community as we think the observed associations might be transferred to hospital staff in other societies.

Reviewer 2:

ABSTRACT

Please, introduce the measurement instruments or questionnaires that you have used to measure the main variables.

Authors:

Thank you very much for this good suggestion. As the number of words of the abstract is restricted to 200, it is not possible to mention all instruments of the questionnaire. Therefore, we added information just on the outcome variable as follows:

The objective of the present study was to improve our understanding of the factors that trigger intentions to leave the profession, which was assessed with the referring scale of the Copenhagen Psychosocial Questionnaire (COPSOQ).

Reviewer 2:

INTRODUCTION:

In the introduction section, please differentiate your hypotheses of the study from the objective, in order to make them clear to the readers.

Authors:

Thank you for the advice to formulate our research questions more precisely.

We inserted the following paragraph:

In view of currently available data, the role of poor quality of care on leaving the profession, taking poor job satisfaction into consideration, is not clear. Therefore, we formulated the following research questions:

  • Is low perceived quality of care associated with intention to leave the profession in young physicians and nurses?
  • To what extent is an existing association between poor quality of care and intention to leave the profession mediated by job satisfaction?

Reviewer 2:

The Lines 78- 81 “This analysis of the associations for young employees could help us to improve …..will continue to function efficiently” should be in the Discussion section, as a practical implications.

Authors:

Thank you for this suggestion.

Since these aspects were our motivation and rationale for the research question, we tend to leave this potential prospect in the introduction part.

Reviewer 2:

Please, finish the introduction section before the methodology, with the aim of the study.

Authors:

Thank you very much for this hint. We removed the information of reanalysis and the according citation from the introduction to the methods part. Additionally we formulated the research question more precisely, see point above.

We inserted the following into the methods section:

Data was taken from a large survey of young physicians and nurses active in acute hospital care and was re-analysed [29].

Reviewer 2:

MATERIALS AND METHODS

I don´t understand how was a randomised cross-sectional study? You say: “In all, 10162 members (6362 physicians, 3800 nurses) were sent an email with an invitation to the online survey”. Please, consider to expand your description of sample accordingly the power analysis used to determine the number of potential individuals that you needed to include in your study.

Authors:

In order to achieve a balanced ratio of nurses and doctors (1:1), it was planned in advance to invite just as many doctors as the total number of nurses of the German Professional Association for Nursing Professions (DBfK) showed, who, according to the member database, fulfilled the inclusion criteria ≤ 35 years and stationary care. A survey was therefore carried out for all members of the DBfK.

Due to the overwhelming number of physicians in the medical societies, a randomized sample was drawn only here among them to match the numbers.

Randomisation was carried out at the study centre using consecutive numbers via the randomisation function in SPSS. The lists were then passed on to the supervisors of the different member databases, who in turn invited the selected members to participate. In one medical society, all members who fulfilled the inclusion criteria were erroneously invited, resulting in an unbalanced ratio to the disadvantage of nurses. A subsequent reduction in the number of participating physicians was waived for statistical reasons.

As we were guided by the total number of members of the DBFK (N=3800) and were heading for a 1:1 ratio, we were at the maximum number of members to invite for this investigation. Therefore, we did not perform a power analysis in the preparation of the study.

Reviewer 2:

Was this sample, representative of the population you wanted to reach? If not, you will have to write it in the limitations section.

Authors:

In the limitations section we listed characteristics referring to the representativeness of the sample. We added the following sentence:

These characteristics limit the representativeness of the sample.

Reviewer 2:

Please, describe what is meaning “COPSOQ”.

Authors:

Thank you for this hint. We now explain the abbreviation in the methods section:

Work-related psychosocial factors were assessed with the Copenhagen Psychosocial Questionnaire (COPSOQ).

Reviewer 2:

I don’t Know how is “tempo at work”.

Authors:

For better understanding we changed the expression “tempo of work” into “working speed” throughout the text.

Reviewer 2:

Maybe, Table 3. Comparison of the constructs job satisfaction and perceived quality of care. Can be more useful in the materials and methods section.

Authors:

Thank you for this suggestion. As this comparison of constructs is a consequence of the interpretation of our results, we think it is more correctly placed in the discussion section.

Reviewer 2:

Figure 1 can be omitted, since the model with its correlations is later offered in Figure 2, which is more complete.

Authors:

Yes, a part of these figures is redundant. Thank you for this advice. We deleted figure 1 in the methods section.

Reviewer 2:

Do not repeat your own results (numbers) in the discussion if not absolutely necessary to make a certain point. Use descriptive language instead.

Authors:

Thank you. We deleted the numbers at the positions where no comparisons to reference data have been drawn:

For this young group of subjects, prevalence values were found to be high for low perceived care, low job satisfaction and intention to leave the profession. For all three variables, the percentage of nurses was significantly greater than for physicians. For the physicians in this study, the mean value for perceived quality of care (xĚ… (SD): 2.4 (0.9)) was less favorable than the 2016 findings for German physicians with their own practices or in hospitals (xĚ… (SD): 1.9 (0.8)) [31]. There are no comparable values with the same scale for nurses.

Reviewer 2:

The discussion fails to clearly attempt to identify/explain reasons for results in this study that differ from other studies looking at similar outcomes.

Authors:

The observed associations of perceived quality of care/job satisfaction are discussed in the paragraph “Associations between perceived quality of care, job satisfaction and intention to leave the profession”. Fact is, that existing literature on this research question for both occupational groups is very sparse. But the observed findings in our study match with these results, as stated in the paragraph.

What differs in our study are the prevalence rates for perceived quality of care, job satisfaction and intention to leave the profession. Referring to this point, we added the following explanation:

(Perceived quality of care in physicians):

Possibly, this difference in perceived quality of care can be explained due to selection bias.

(job satisfaction and intention to leave the profession):

With regard to the comparison to COPSOQ data of employees aged up to 34 years, no changes in the mean values of job satisfaction and intention to leave the profession could be observed in a sensitivity analysis when 35-year-olds (N=50) were excluded. To what extent the differences in the mean values can be explained by working conditions that possibly deteriorate over time or by a selection bias cannot be answered in this study.

Reviewer 2:

CONCLUSION. The conclusion should state your findings. The last sentences are not conclusions of your research.

Authors:

Thank you for this comment. We changed the conclusion section as follows:

The results of this study point out that – for young hospital employees – both the low perceived quality of care and low job satisfaction are associated with intention to leave the profession. Perceived quality of care is mainly directly associated with intention to leave the profession, although there is also partial association mediated through job satisfaction. Further studies in longitudinal design are needed to confirm these findings.

Reviewer 2:

REFERENCES. Please review some mistakes in the references.

The numbers of the final references do not match the references in the text. See the references section (line 351):

Authors:

Thank you very much for this hint!

We corrected the reference section properly.

Round 2

Reviewer 2 Report

Thanks for your careful revisions of the manuscript. The manuscript is by now much clearer.